# Potential Roles of Anti-Inflammatory Plant-Derived Bioactive Compounds Targeting Inflammation in Microvascular Complications of Diabetes

**DOI:** 10.3390/molecules27217352

**Published:** 2022-10-29

**Authors:** Yahia A. Kaabi

**Affiliations:** Medical Laboratory Technology Department, Faculty of Applied Medical Sciences, Medical Research Center, Jazan University, Jazan 42200, Saudi Arabia; ykaabi@jazanu.edu.sa; Tel.: +966-549918001

**Keywords:** diabetes mellitus, inflammation, diabetic nephropathy, diabetic retinopathy, diabetic neuropathy, bioactive compounds

## Abstract

Diabetes mellitus (DM) is a group of metabolic disorders, the characteristics of which include chronic hyperglycemia owing to defects in insulin function, insulin secretion, or both. Inflammation plays a crucial role in DM pathogenesis and innate immunity in the development of microvascular complications of diabetes. In addition, hyperglycemia and DM mediate a proinflammatory microenvironment that can result in various microvascular complications, including diabetic nephropathy (DNP), diabetic neuropathy (DN), and diabetic retinopathy (DR). DNP is a major cause of end-stage renal disease. DNP can lead to albuminuria, decreased filtration, mesangium expansion, thickening of the basement membrane, and eventually renal failure. Furthermore, inflammatory cells can accumulate in the interstitium and glomeruli to deteriorate DNP. DN is another most prevalent microvascular complication of DM and the main cause of high mortality, disability, and a poor quality of life. DNs have a wide range of clinical manifestations because of the types of fiber dysfunctions and complex structures of the peripheral nervous system. DR is also a microvascular and multifactorial disease, as well as a major cause of visual impairment globally. Pathogenesis of DR is yet to be fully revealed, however, numerous studies have already confirmed the role of inflammation in the onset and advancement of DR. Despite evidence, and better knowledge regarding the pathogenesis of these microvascular complications of diabetes, there is still a deficiency of effective therapies. Bioactive compounds are mainly derived from plants, and these molecules have promising therapeutic potential. In this review, evidence and molecular mechanisms regarding the role of inflammation in various microvascular complications of diabetes including DNP, DN, and DR, have been summarized. The therapeutic potential of several bioactive compounds derived from plants in the treatment of these microvascular complications of diabetes has also been discussed.

## 1. Introduction

Diabetes mellitus (DM) is a major public health problem and is one of the top 10 causes of death worldwide [1]. The global burden of DM has increased markedly and is likely to continue to increase significantly in the upcoming decades. Unfortunately, nearly half (46.2%) of diabetes-linked deaths occur in individuals aged below 60 years [2]. DM is a chronic and metabolic disease, which is characterized by increased levels of plasma glucose [3]. In addition, low-grade inflammation is a characteristic of DM, which mediates hyperglycemia, insulin resistance, and the development of multiple complications [4,5,6]. Growing evidence suggests the role of inflammation in DM pathogenesis and targeting inflammation could be an effective approach in controlling and preventing DM [7]. In addition to this, the chronic hyperglycemia of DM can lead to various vascular complications, which can take place both in small and large blood vessels. Various mechanisms, including inflammation (Figure 1), can contribute to the development of vascular complications of DM [8]. It has been revealed that DM can mediate pathognomonic alterations in the microvasculature, which can affect the capillary basement membrane (CBM), including arterioles in the muscle, skin, myocardium, retina, and glomeruli via increasing their thickness, which can further result in diabetic microangiopathy development [9]. The microvascular complications of diabetes, including neuropathy, retinopathy, and nephropathy, can cause dependency and disability as well as inducing morbidity and mortality [10].

Diabetic neuropathy (DN) is another commonly observed microvascular complication of DM, that affects as high as 50% of individuals with type 1 diabetes (T1DM) and type 2 diabetes (T2DM) [11]. DN characteristics include low-grade inflammation [12]. Furthermore, various proinflammatory cytokines (CKs), including C-reactive protein (CRP), monocyte chemoattractant protein-1, interleukin (IL)-1, IL-6, IL-8, and tumor necrosis factor alpha (TNF-α), are primarily generated via activated immune cells. However, they are also generated via resident adipocytes and macrophages, and play a major role in inducing inflammation. Locally generated and circulating E-selectin, vascular cell adhesion molecule-1, and intracellular adhesion molecule-1 (ICAM-1), suggest the presence of low-grade chronic inflammation, which has been linked with complications of DM [13,14]. Moreover, chemokines secreted from damaged/infected tissues induce the endothelium to raise the expressions of chemokines and adhesion molecules [15].

Like DN, diabetic retinopathy (DR) is also a vital complication in the case of diabetes mellitus [16]. In working-age people globally, DR is a major cause of preventable blindness [17]. Around one-third of diabetic patients suffer from DR and 10% have sight-threatening complications, including proliferative DR and diabetic macular edema [17,18]. DR also involves low-grade inflammation, including a cascade of adhesion molecules and inflammatory mediators [19,20]. It has been observed that subclinical chronic inflammation can result in the occurrence of diabetes and diabetes-linked long-term complications, including DR [21]. Therefore, understanding the inflammatory mechanisms might be beneficial in developing strategies and therapeutic agents to control or even prevent diabetes and related complications, before they result in irreversible organ damage [21,22]. Diabetic nephropathy (DNP), also known as diabetic kidney disease, denotes the weakened kidney functions observed in DM patients. DNP progresses in several stages and is associated with blood pressure and glycemic control [23]. Indeed, the occurrence of DNP significantly increases with the occurrence of diabetes worldwide [2].

DNP is typically considered as a nonimmune disease, however, growing evidence suggests that inflammatory and immunologic pathways have a significant contribution in its progression and development [24,25]. DNP processes include: nuclear factor-κB (NF-κB) [26,27]; various growth factors (transforming growth factor beta (TGF-β); insulin-like growth factor (IGF) [28,29,30]; growth hormone, vascular endothelial growth factor (VEGF)) [28,29,30]; multiple enzymes (including nitric oxide synthase and cyclooxygenase-2) [31,32,33,34]; adhesion molecules (ICAM-1) [35,36]; chemokines (monocyte chemoattractant protein-1) [37]; and a number of cells (such as- macrophages, monocytes, and leukocytes) [38,39,40]. Currently available oral antidiabetic medications include metformin, glucagon like peptide 1 (GLP-1) receptor agonists, dopamine agonists, dipeptidyl peptidase IV inhibitors, alpha-glucosidase inhibitors, thiazolidinediones, meglitinides, and sulfonylureas [41,42,43]. However, there are numerous complications linked with the use of available oral antidiabetic agents and insulin therapy, including various unwanted side effects (for example, hypothyroidism, tachycardia, weight gain, hepatic failure, and lactic acidosis), limited drug tolerability, and cost [44,45,46]. These complications have triggered the search for alternative medicines with fewer side effects, as well as improved potency and efficacy. There is a long history of using plants as medicines in the treatment of many diseases. In addition, plant-derived bioactive compounds show better patient acceptance and tolerance, as well as having a long history of clinical use [47,48]. There is growing evidence regarding natural bioactive compounds, since they are cheaper alternatives than synthetic drugs, exert fewer toxic side effects, and have great potential in treating and preventing various diabetes-linked microvascular complications [49]. In this review, the evidence regarding the mechanistic link between inflammation and the various microvascular complications of diabetes, including DNP, DN, and DR, have been summarized. Moreover, the therapeutic potential of several bioactive compounds derived from plants in the management of these microvascular complications of diabetes has also been discussed.

## 2. The Role of Inflammation in Diabetes Mellitus

Inflammation plays a crucial role in metabolic dysregulation. The inhibition of inflammation is considered as a metabolically protective process that decreases the development of T2DM and insulin resistance [50]. The insulin resistance of skeletal muscles, adipose tissues, and the liver, induces insulin secretion from the pancreas, which further maintains a normal blood sugar level [51,52,53]. The effective link between the target tissues of insulin (such as skeletal muscles, liver, adipose tissues, and pancreas) and insulin-secreting cells, maintain metabolic homeostasis in regard to physiological alterations that take place in lipemia or glycemia due to starvation or food consumption. Insulin resistance can cause the partial disruption of communication between these tissues, where insulin target tissues show resistance towards the insulin signaling mechanism in spite of the initial compensation mediated by the pancreas. These events are more prominently observed in the case of T2DM, where there is a deficiency of sufficient insulin production to regulate blood glucose levels. Interestingly, these target tissues have their own specialized macrophages to keep tissue integrity and support crucial physiological activities. In addition, at each stage of the development of T2DM, the number of macrophages in these tissues goes through alteration [50,54,55].

Tissue macrophages are known as very strong mediators of insulin resistance, sensitivity, and signaling. Macrophages can rapidly adapt their activities and respond to environmental signals. The endoplasmic reticulum (ER) plays a crucial role in the insulin resistance of cells. ER stress can take place because of the disturbance of homeostasis, which is a major sign of metabolic disorders [56]. In obesity, continuous metabolic pressure can disturb important ER activities, which can further cause metabolic collapse, inflammation, and weakened cellular health [56]. Furthermore, hyperglycemia can significantly induce reactive oxygen species (ROS), which can further lead to an elevated level of activation of inflammatory signaling pathways [50]. T2DM and T2DM-linked cardiovascular complications, including macrophages/monocytes and atherosclerosis, have significant contributions in regulating inflammation. It has been demonstrated that the processes of weakened insulin function are linked with the phosphorylation of serine/threonine, which can mediate the signaling of insulin receptor (IR) [57]. Furthermore, phosphorylation can be inhibited via various proinflammatory CKs, including IL-6, IL-1β, and TNFα secreted by macrophages. In addition, these CKs can activate several serine kinases, including the mammalian target of rapamycin 32 (mTOR32), ribosomal protein S6 kinase, c-Jun N-terminal kinase (JNK), and IκB kinase β (IKKβ) in adipocytes, which further facilitate the suppressive phosphorylation of insulin receptor substrate 1 to cause insulin resistance [50,58].

Moreover, the same kinases can significantly trigger immune response via the toll-like receptor (TLR), which, following activation, further induces the generation of multiple CKs [59,60]. Adipose tissues are composed of vascular cells, fibroblasts, macrophages, lymphocytes, and preadipocytes. An increased level of macrophages was observed in visceral adipose tissues as compared to subcutaneous adipose tissue, which further suggests that the buildup of visceral fat can result in insulin resistance as well as metabolic diseases. Obesity can also alter the cellular composition of adipose tissues and the activation of immune cells [61]. Elevated cell death, hypoxia, and adipocyte hypertrophy because of the elevated buildup of lipids (mainly triglycerides), can trigger the release of various proinflammatory molecules including adipokines, MCP-1, IL-8, IL-6, and TNF-α via immune cells (mainly macrophages) and adipocytes, which can eventually elevate infiltration of circulating immune cells and monocytes into adipose tissues [62,63]. In addition, recruited monocytes were found to be differentiated into proinflammatory M1 macrophages, which can further disturb the balance between M1 and M2 macrophages and decrease anti-inflammatory responses from M2 macrophages. Moreover, this event can markedly trigger the release of various proinflammatory adipokines and CKs, which can ultimately result in decreased glucose tolerance and dysfunction of adipose tissues [50,64].

Interestingly, most of the proinflammatory stimuli can cause activation of both IKKβ and JNK signaling pathways. Various proinflammatory CKs, including IL-1β and TNF-α, can activate IKKβ/NF-κB and JNK pathways via both receptor-mediated and non-receptor processes via activation of several receptors, including TLR and glycation end products receptor (RAGE). Furthermore, JNK can mediate insulin resistance via the phosphorylation of serine residues in IKKβ and insulin receptor substrate 1, which can further mediate insulin resistance via NF-κB. It has been revealed that the activation of NF-κB and JNK signaling pathways can result in the generation of various proinflammatory mediators and CKs, which can cause activation of these signaling pathways through feed-forward processes [65]. Dysfunctional adipose tissues and obesity have significant contributions in the pathogenesis of insulin resistance and various liver diseases, including non-alcoholic fatty liver disease (NAFLD) and non-alcoholic steatohepatitis (NASH) [66]. The link between T2DM and NAFLD is bidirectional and very complex. NAFLD plays an important role in DM development. DM can also play an important role in the advancement of hepatocellular carcinoma, liver cirrhosis, NAFLD, and NASH [50,67,68].

## 3. The Role of Inflammation in Microvascular Complications of Diabetes

### 3.1. Diabetic Neuropathy (DN)

Diabetic neuropathy (DN) is another major microvascular complication of diabetes that affects both autonomic and peripheral nerves [69]. It has been observed that the risk of DN development is directly linked with the extent and duration of hyperglycemia (Figure 2). Some individuals are also prone to the development of such complications owing to genetic predisposition [9,70]. The exact cause of hyperglycemia-mediated damage to the peripheral nerves is yet to be confirmed, however, numerous studies have confirmed the role of inflammation in the development of DN [71,72,73,74,75,76,77]. The activation of the IKKβ/NF-κB axis plays a crucial role in inflammatory responses. Various stimuli include proinflammatory CKs, oxidative stress (OS), and hyperglycemia [5,78]. Many studies have demonstrated that NF-κB axis activation can induce immune and inflammatory responses, which might further result in the expression of CKs and adhesion molecules, and cellular injury [79,80,81]. In a streptozotocin (STZ)-induced diabetic model, ischemia-reperfusion injury triggered overexpression of NF-κB in the Schwann cells and the diabetes-induced activation of sciatic nerve endothelial cells, along with the subsequent elevated level of ICAM-1 expression and the infiltration of monocyte macrophages as compared to controls, which further indicates that the NF-κB activation mediated the rise in the level of inflammation in diabetic nerves [82]. In addition to this, the NF-κB signaling pathway controls the expressions of a range of inflammatory genes and their downstream functions. The NF-κB signaling axis was found to regulate the activities of many genes, including cyclooxygenase-2 (COX-2), nitric oxide synthase (NOS), lipoxygenase, and endothelin-1 in the animal models of DN [71]. TNF-α also mediated the activation of mitogen-activated protein kinase (MAPK) and the overexpression of COX-2. These processes are also associated with DM-mediated proinflammatory responses and neuropathic changes [72].

The increased levels of TNF-α were found in the sciatic nerves of diabetic rat and mouse models. This event was associated with the dysfunctions of large and small nerve fibers, as confirmed by the reduced levels of motor nerve conduction velocities (NCV) and the density of intraepidermal nerve fiber in the animal models of diabetes [72]. These alterations were averted via treatment with a COX-2-selective inhibitor or inactivation of the COX-2 gene [72]. The suppression of 12/15 lipoxygenase ameliorated mechanical allodynia, as well as sensory and motor NCV in another animal model of DM; however, this suppression did not alter thermal hypoalgesia [71]. When the mouse models were experiencing pain, the upregulation of several inflammatory mediators including TNF-α, inducible NOS (iNOS), and COX-2 in sensory neurons of db/db mouse models during the early stage of DN, was observed [83]. In an experimental model of T2DM DN, blocking the raised level of these inflammatory markers also blocked pain, which further indicates that inflammatory response in sensory neurons can cause pain [83]. Nevertheless, the activation of the inflammatory signaling mechanisms with the release of CKs and downstream signaling might show dual activities. IL-6 was found to be co-secreted with TNF-α and is regarded as a proinflammatory cytokine [84]. Multiple studies have revealed that the administration of pharmacological doses of IL-6 ameliorated sensory and motor NCV in diabetic rat models, which also mediated raised the level of nerve blood flow and corrected the altered tactile allodynia and thermal nociception [71,85,86]. However, more studies, particularly clinical studies, are required to confirm these conflicting findings regarding IL-6 [5].

Inflammation has also been linked with the abnormality of several heat-shock proteins that provide protection to cells against environmental stress and play a role as molecular chaperones. It has been revealed by in vitro studies that Hsp70 (70 kDa heat shock protein) bound with high affinity to the plasma membrane caused the activation of NF-κB, as well as upregulating the expressions of IL-6, IL-1β, and TNF-α in human monocytes [5,87,88,89]. DM also significantly increased inflammatory pathways, as confirmed by the transcriptomic studies of sensory neuron RNA derived from diabetic Hsp70 knock-out and wild-type mice utilizing RNA sequencing [90,91]. In the animal models of T1DM, the modulation of Hsp90 and Hsp70 by using targeted small molecules improved bioenergetics, morphologic, electrophysiologic, and psychosensory deficits of DPN [90,91]. The concentration of circulating HSP27 was also found to be linked with distal sensorimotor neuropathy [92]. It has been reported that hyperglycemia, insulin resistance, and the loss of insulin signaling, as well as dysregulations in dyslipidemia and lipid metabolism, resulted in systemic inflammation and dangerous cycles of nitrosative/oxidative stress, the buildup of cellular injury, mitochondrial, and endoplasmic stress [12,93,94,95,96,97,98]. Lipotoxicity, insulinopenia, and glucotoxicity, generated nitrosative/neuronal stress and activated several downstream kinases including jun N-terminal kinase (JNK), MAPK, protein kinase C (PKC), and redox-sensitive transcriptional factors including NF-κB. Moreover, these factors have a significant contribution in inducing the generation of various chemokines and CKs, including chemokine (C-X-C motif) ligand 1 (CXCL1), chemokine (C-C motif) ligand 2 (CCL2), TNF-α, IL-8, IL-6, IL-2, and IL-1β [96,97]. Chemokines and CKs also enhanced existing immune responses and inflammation, as well as mediated the activation of various downstream cellular nitrosative stress/OS, which can further mediate more neuronal injury [96].

The aforesaid pre-clinical results were also observed in human studies. A study involving 55 healthy controls and 150 volunteers with DM (with and without DN) reported the elevated serum levels of various inflammatory CKs (such as CRP and TNF-α), as well as endothelial dysfunction markers in DN subjects [74]. In addition, these markers were also elevated in individuals with painful DN [74]. This finding was also observed in a different study, where an elevated level of IL-2 in individuals with neuropathic pain was observed [99]. The increased serum levels of IL-10 and IL-6 were reported in individuals with prediabetes and T2DM linked with signs of motor nerve demyelination, as well as markers of motor axonal and large nerve fiber sensory injury [100]. IL-10 negatively controls TNF-α and plays a role as an anti-inflammatory CK; therefore, the elevated levels of IL-10 may take place as part of a compensatory mechanism.

In another cohort study involving 1000 individuals, it was observed that the serum levels of various inflammatory CKs, including IL-6 and IL-1β, were positively linked with measures of peripheral DN in sex- and age-adjusted analyses [101]. The role of inflammation in the development of autonomic dysfunction (particularly cardiovascular autonomic neuropathy (CAN)) has also been confirmed. As compared to T1DM individuals without CAN, the markedly increased levels of inflammation markers (such as increased TNF-α levels) were also confirmed in individuals with CAN and T1DM [77]. In a case-control study, the IL-6 level was found to be negatively correlated with CAN measures, whereas the ratio of adiponectin to leptin was positively correlated with CAN measures [73]. In individuals with progressive DN, the gene expression patterns of sural nerves were extremely functionally enriched in immune response and inflammatory pathways, as well as specifically up-regulated genes including various CKs (IL-6, IL-2, and IL-1β) and chemokines (CXCL1 and CCL2) [75]. On the other hand, the gene expression patterns of sural nerves with the anatomical evidence of regeneration from DN individuals who have stable disease, showed a marked downregulation of genes and pathways linked with the immune response and inflammation [76].

### 3.2. Diabetic Retinopathy (DR)

Diabetic retinopathy (DR) is a common complication of DM and a microvascular disease [102]. DR diagnosis is carried out by the clinical manifestations of retinal vascular abnormalities [102]. It has been observed in T2DM that persistent hyperglycemia can result in proinflammatory diabetic milieu in the retina [103]. Furthermore, this aberrant metabolic condition is reflected by cellular hypoxia, OS, the accumulation of free radicals, altered metabolic pathways, thrombophilia, and altered platelet physiology and functions [104,105]. Chronic persistent hyperglycemia can induce proinflammatory conditions where multiple pathways such as PKC, OS, advanced glycation end products, and the accumulation of polyol, can induce the expression of several inflammation mediators [106]. These stimuli can further disturb certain pathways or specific cells within the vascular and cellular domains of the retina to induce inflammatory cascades. After stimulation via these DM by-products, pericytes, retinal ganglion cells, inner retina (primarily the inner nuclear layer), endothelial cells, retinal pigment epithelium, Müller cells, and diversified microglial cells accompanied by the constituents of the inner blood-retinal barrier, play a role as the major source of inflammation [106]. Gliosis or glial activation was found to occur in individuals with no clinical signs of DR. In these patients, elevated GFAP levels were also observed in the aqueous humor [107]. Müller cells are the main type of glial cell in the retina [108]. These glial cells also serve as the major source of many factors including inflammatory modulators, which indicates that the activation of retinal glial cells can lead to the onset of the inflammatory mechanism accountable for retinal injury at later stages of DR.

The increased levels of several inflammatory CKs, including MCP-1, TNF-α, IL-8, IL-6, and IL-1β, have been observed in the ocular tissues of non-proliferative DR (NPDR) individuals. In addition, the increased levels of TNF-α and IL-8 have been observed in the eyes of NPDR patients more than in the individuals with active PDR [109]. Elevated levels of these CKs are generated via activated microglia, macroglia, endothelial cells, and at a later stage even neurons, which further indicates the elevated function of these inflammatory CKs in early DR stages and the advancement of inflammation in all types of retinal cells [110]. In DM, the buildup of these inflammatory mediators can result in the early death of neurons in the retina. In experimental ischemic mouse models, several CKs including IL-3, IL-1, and macrophage inflammatory protein 1, played significant roles in angiogenesis, which indicates that inflammation precedes and contributes to neovascularization development in the case of PDR [111,112,113]. The increased vitreous concentrations of several neurotrophins (NTs), including glial cell line-derived neurotrophic factor, ciliary neurotrophic factor, NT-4, NT-3, brain-derived neurotrophic factor, and nerve growth factor, were detected in patients with DR, along with elevated NPDR levels, than in PDR [109]. These aforesaid events indicate that glial cells, including MGCs, may try to salvage compromised neurons during the early stage of DR. Furthermore, the elevated vitreous levels of these NTs and CKs in consort with various other growth factors, including hepatocyte growth factor (HGF), basic fibroblast growth factor, IGF-1, platelet-derived growth factor, and VEGF, have been detected in PDR individuals [114]. The investigation of vitreous samples obtained from patients with PDR showed elevated concentrations of soluble cytokine receptors including soluble IL-2 receptors [115].

This rise can lead to a known negative regulatory process of the cytokine signaling pathway, which indicates that the counter-regulatory processes of inflammation and angiogenesis are present within the eye. PDR development is a multistage process (such as angiogenesis), where the degradation of the basement membrane, cell migration, the proliferation of endothelial cells, and subsequently the formation of capillary tubes, can take place. Matrix metalloproteinases (MMPs) are the crucial controllers of the aforesaid tissue remodeling and migratory mechanisms. An increased level of MMPs was also observed in PDR [116]. Moreover, the increased levels of VEGF, inflammatory CKs, and angiopoietin-2 (an angiogenesis modulator), were also observed in patients with diabetic macular edema [117,118]. There are no established biomarkers to monitor the severity of DR, however, various studies have revealed that intravitreal level of MCP-1, IL-6, HGF, and VEGF, increases with the advancement of DR from NPDR to active PDR [114,115]. The levels of IL-6 were also found to be positively correlated with retinal macular thickness [119].

Various aforesaid inflammatory mediators are found to be activated in the case of DR, however, the signaling pathway associated with initiating this response is unclear. A complication of utilizing aqueous or vitreous levels to evaluate the expressions of proteins linked with DM and DR, is that the observed alterations may only show alterations in circulating serum concentrations. Interestingly, the total content of vitreous proteins was found to be the same in PDR and NPDR individuals [109], which indicates that the elevated levels of vitreous proteins exhibit an increased level of release instead of just the release of serum proteins into the vitreous, owing to a disturbed blood-retinal barrier. However, there are numerous features of inflammatory responses including tissue edema, vascular permeability, elevated blood flow, the upregulation of CKs, complement and microglial activation, macrophage and neutrophil infiltrations, and leukostasis, in experimental animal models and human DR patients [120,121,122].

### 3.3. Diabetic Nephropathy (DNP)

Diabetic nephropathy (DNP) or diabetic kidney disease is a microvascular complication of DM, which is responsible for end-stage kidney disease [123]. DNP characteristics include the loss of glomerular filtration rate, diabetic glomerular lesions, and abnormal urine albumin excretion [124]. Numerous experimental and clinical studies have already confirmed the presence of inflammation in DNP. It has been revealed that individuals with T2DM and overt nephropathy show a substantial level of various inflammatory markers, such as IL-6, fibrinogen, serum amyloid A (SAA), and CRP [125,126]. The increased levels of IL-6, SAA, and CRP, were also observed in individuals with increased thickness of glomerular basement membrane (GBM). A link between acute phase markers (including IL-6 and fibrinogen) and GBM thickening (a major lesion of diabetic glomerulopathy) has also been confirmed. In a study, an elevated ICAM-1 level was observed in db/db mouse models, which mediated inflammatory response via elevating leukocyte adherence and infiltration in tubules and glomeruli, accompanied by an elevated level of macrophage infiltration [36]. Collectively, these results strongly suggest that ICAM-1 mediates renal damage in diabetes. In another study, Kelly et al. [127] confirmed in a model of hypertension and DM that in spite of increased blood pressure and hyperglycemia, treatment with ruboxistaurin (a PKC-β inhibitor) preserved the renal functions and decreased albuminuria in rats. A range of PKC isoforms is activated in diabetes and signals various cellular responses, such as the expression and activation of the mediators of inflammation, including proinflammatory CKs [128].

The incubation of macrophages with GBMs derived from diabetic rat models showed increased TNF-α and IL-1 levels, as compared to macrophages incubated with GBMs derived from normal rats [128,129]. In addition, there is a link between proinflammatory CKs and DNP. Numerous studies have demonstrated that IL-1 increases adhesion molecules in glomerular endothelium and expressions of these molecules in other areas of the kidney [130]. Renal tubular epithelium and mesangial cells overexpress E-selectin (an adhesion receptor) and ICAM-1. Furthermore, IL-1 triggers the generation of prostaglandin E2 in mesangial cells, which further alters glomerular hemodynamics [130]. IL-1 also induces hyaluronan production to cause cell proliferation in patients with DM, which can eventually result in DNP development. The increased levels of IL-1 and the accumulation of macrophages have also been observed in experimental models with albuminuria [130]. IL-1 was also found to increase the expression of chemokines and modify vascular permeability, which further resulted in the synthesis and proliferation of extracellular matrix in mesangium [131,132].

IL-6 is another CK that exerts pleiotropic effects in DNP. Numerous studies have already demonstrated an increased level of IL-6 in DNP. It has been reported that IL-6 exerts direct action in infiltrating and glomerular cells, which altered the dynamics of the extracellular matrix affecting GBM thickening [130,133]. IL-6 also affects vascular permeability, extracellular matrix overexpression, and increased proliferation [130]. An increased level of serum IL-6 was also observed in individuals with T2DM and DNP [134,135]. Moreover, IL-6 plays various other roles including upregulating ICAM-1 and increasing the generation of other inflammatory CKs via mesangial cells. In DNP, the elevated levels of IL-6 and various other interleukins have harmful effects on endothelium apoptosis [130]. In the diabetic kidney, IL-18 (a CK) has several sources as infiltrating, monocytes, macrophages, T-lymphocytes, and proximal tubule cells. A direct link between albumin excretion rate, albuminuria, and IL-18 has already been observed, which further suggests its role in DNP [131,136]. In inflammation, IL-18 also exerts various harmful effects via various tissues and pleiotropic activities. Interestingly, TNF-α was also found to be generated via infiltrating cells, such as T lymphocytes, macrophages, monocytes, and kidney cells. Various findings suggest that TNF-α can be deposited as a proactive form and its harmful effects have already been demonstrated in humans and experimental models [130]. These harmful effects have been demonstrated as DM nephropathy, glomerulonephritis, nephritis, and hypertension. In a study, a link between albumin excretion and renal TNF-α has been reported by Navarro-González et al. [137] in diabetic mouse models. This finding further suggests that TNF-α plays role in DNP pathogenesis by causing tissue and cell damage. A link between albuminuria and enhanced trigger for TNF-α overexpression has also been observed [131,138].

## 4. Therapeutic Potential of Anti-Inflammatory Bioactive Agents in the Treatment of Microvascular Complications of Diabetes

### 4.1. Curcumin

Curcumin (Figure 3) is present in the rhizomes and roots of Curcuma longa (turmeric) (Table 1) [139]. Curcumin has a range of biological properties including anti-tumor, anti-oxidant, and anti-inflammatory properties [140]. Among these properties, the anti-inflammatory effects of curcumin are regarded as the basis of its therapeutic effects in treating various diseases [139]. In a study, Gupta et al. [141] observed that curcumin exerted a substantial hypoglycemic effect as compared to the diabetic group. Levels of retinal glutathione were reduced by 1.5-fold, while catalase, superoxide dismutase, and antioxidant enzymes showed over 2-fold reduction in function in the diabetic group, while curcumin positively controlled the antioxidant system (Table 2). Curcumin also prevented the over 2-fold rise in the levels of VEGF, TNF-α, and proinflammatory CKs. Increased thickness of CBM and degenerated endothelial cell organelles were observed through transmission electron microscopy in the diabetic retina; however, curcumin averted the rise in thickness of CBM and structural degeneration in the retina of diabetic rat models. Collectively, these findings suggest that curcumin has the potential in the prevention of DR [141].

In another study, Sun et al. [142] observed that curcumin ameliorated in vivo fibrosis and histological irregularities of a diabetic kidney. Furthermore, curcumin suppressed expressions of renal inflammatory genes and decreased the phosphorylation of caveolin-1 (cav-1) at Tyr^14^, as well as the toll-like receptor 4 (TLR4) expressions. Following the pretreatment of podocytes with curcumin, it was observed that curcumin decreased high glucose (HG)-induced cav-1 phosphorylation, the generation of proinflammatory CKs, and TLR4. In rat kidneys, immunohistochemistry studies revealed that the increased expression of TLR4 is more evident in the renal interstitium as compared to the glomerulus where podocytes are found. Furthermore, in the kidney, curcumin-mediated anti-inflammatory properties on other cells might be mediated via the same molecular mechanisms as in podocytes [142].

**Table 1 molecules-27-07352-t001:** Classification and major dietary sources of anti-inflammatory bioactive compounds that have the potential to treat microvascular complications of diabetes.

Bioactive Compound	Compound Group	Dietary Sources	References
Curcumin	Polyphenol	Turmeric	[139]
Epigallocatechin-3-gallate (EGCG)	Polyphenol	Green tea, hazelnuts, avocados, apples, pears, cherries, kiwis, blackberries, strawberries, and cranberries	[143,144,145]
Resveratrol	Polyphenol	Peanuts, grapes, and peanut sprouts	[146]
Genistein	Isoflavone	Legumes and soy products	[147]
Berberine	Alkaloid	*Tree turmeric, barberry, oregon grape,* *and goldenseal*	[148]
Quercetin	Flavonoid	Citrus fruits, broccoli, cherries, berries, grapes, and onions	[149]
Naringin	Flavonoid glycoside	Skin of oranges and grapefruit	[150]
Apigenin	Flavonoid	Wheat sprouts, chamomile, tea, oranges, onions, and parsley	[151]
Kaempferol	Polyphenol	Berries, asparagus, onion, tarragon, chives, kaledill, and spinach	[152]
Baicalein	Flavonoid	Root of huangqin	[153]
Eriodictyol	Flavonoid	Citrus fruits	[154]

**Table 2 molecules-27-07352-t002:** Anti-inflammatory effects of bioactive compounds in animal models of diabetic macrovascular complications.

Bioactive Compound	Study Model	Duration	Study Outcomes	References
Curcumin	Rat model of diabetic retinopathy	16 weeks	Retinal glutathione was reduced by 1.5-fold, while catalase, superoxide dismutase, and antioxidant enzymes showed over 2-fold reduction in function in the diabetic group; positively controlled the antioxidant system; prevented the over 2-fold rise in the levels of vascular endothelial growth factor (VEGF), tumor necrosis factor alpha (TNF-α), and proinflammatory cytokines (CKs)	[141]
	Rat model of diabetic nephropathy	12 weeks	Inhibited expressions of renal inflammatory genes and reduced phosphorylation of caveolin-1 at Tyr^14^ as well as the toll-like receptor 4 (TLR4) expression; reduced high glucose (HG)-induced caveolin-1 phosphorylation, generation of proinflammatory CKs, and TLR4	[142]
Epigallocatechin-3-gallate (EGCG)	Rat model of diabetic nephropathy	50 days	Inhibited lipid peroxidation, proteinuria, and hyperglycemia	[155]
	Rat model of diabetic nephropathy	24 weeks	Reduced the expressions of vascular cell adhesion molecule-1 (VCAM-1) and intracellular adhesion molecule-1 (ICAM-1); improved glomerular enlargement and mesangial matrix expansion	[156]
Resveratrol	Mouse model of diabetic nephropathy	28 days	Reduced expressions of NADPH Oxidase 4 (NOX4), NF-_K_B (P65), and RAGE), 24 h urinary microalbumin quantitative, serum creatinine (SCr), and blood urea nitrogen level; improved renal pathological structure	[157]
	Rat model of diabetic nephropathy	7 days	Significantly decreased protein carbonyl oxidative stress (OS) markers and superoxide anions; markedly decreased protein phosphorylation and expression of adenosine monophosphate-activated protein kinase (AMPK) protein	[158]
Genistein	Rat model of diabetic nephropathy	6 weeks	Improved kidney functions as well as decreased levels of blood glucose and SCr; downregulated expressions of p53, p65, mitogen-activated protein kinase (MAPK), and NOX4; elevated mitochondrial membrane potential; protected podocyte integrity; decreased OS and expansion of the mesangial matrix	[159]
	Rat model of diabetic retinopathy	2 weeks	Significantly inhibited TNF-α mRNA and protein levels; inhibited TNF-α secretion and phosphorylation of P38 and extracellular signal-regulated kinase (ERK) in activated microglial cells via suppressing tyrosine kinase; interfered with various inflammatory signaling pathways including P38 MAPKs and ERK in activated microglia	[160]
Berberine	Rat model of diabetic retinopathy	8 weeks	Deactivated the nuclear factor-κB (NF-κB) signaling; reduced diabetes-mediated OS; decreased apoptosis of the retinal ganglion cell layer; reversed HG-mediated effects on OS and apoptosis in Müller cells	[161]
	Rat model of diabetic retinopathy	2 weeks	Suppressed the expressions of transient receptor potential vanilloid 1 and protein kinase C epsilon might be activated via hyperglycemia-induced inflammation	[162]
Quercetin	Rat model of diabetic peripheral neuropathy	6 weeks	Significantly improved the raised levels of interleukin-1β and TNF-α; markedly reduced the expressions of NF-κB, myeloid differentiation factor 88, and TLR4	[163]
Naringin	Rat model of diabetic retinopathy	12 weeks	Suppressed NF-κB signaling	[164]
	Rat model of diabetic nephropathy	12 weeks	Reduced OS; inhibited NOX4 expressions	[165]
Apigenin	Rat model of diabetic nephropathy	7 days	Regulated the expressions of miR-423-5p-USF2 axis; suppressed ROS and Nox4	[166]
Baicalein	Mouse model of diabetic nephropathy	12 weeks	Reduced inflammation and OS via modulating MAPK and Nrf2 signaling	[167]
Eriodictyol	Rat model of diabetic retinopathy	10 days	Markedly decreased the levels of endothelial nitric oxide synthase, VEGF, ICAM-1, and TNF-α in a dose-dependent manner	[168]

### 4.2. Epigallocatechin-3-Gallate (EGCG)

Epigallocatechin-3-gallate (EGCG) is the most abundant and biologically active catechin present in green tea [145]. EGCG showed anti-inflammatory effects in numerous chronic kidney disease models including DNP, renal fibrosis, and immune-mediated kidney diseases. EGCG typically shows anti-inflammatory properties via suppressing the NF-κB signaling pathway, which further leads to the reduced recruitment of inflammatory cells and the decreased stimulation of proinflammatory mediators. It has been revealed by an in vitro study of AGE-mediated inflammation of HEK cells and human mesangial cells, that these renal cells markedly elevated the level of proinflammatory CKs, including TNF-α, following advanced glycation end products (AGE) treatment; however, EGCG pretreatment may decrease such elevation [169]. Furthermore, the suppressive effect of EGCG was found to be similar to L-165041, a potent agonist of the peroxisome proliferator-activated receptor delta (PPARδ) receptor [169]. L-165041 and EGCG also stimulated PPARδ and inhibited the expression of receptor for AGE. EGCG also inhibited the NF-κB activation by reducing the expressions of nuclear NF-κB and IκB in the cytoplasm [169]. Nonetheless, this event was obliterated via siRNA specific to PPARδ.

Collectively, these results suggest that the suppressive effect of EGCG on AGE-mediated inflammation was mediated via the activation of PPARδ [169]. Treatment with the combination of alpha-lipoic acid and EGCG exerted anti-inflammatory properties on HG-treated HEK cells via decreasing the expression of RAGE and the generation of various proinflammatory CKs, including IL-6 and TNF-α [170]. EGCG also showed anti-inflammatory properties in a rat model of STZ-induced DNP [155]. Treatment with EGCG markedly decreased the NF-κB signaling pathway, which further resulted in reduced expressions of p-IκB-α, NF-κB, and various downstream enzymes including iNOS and cyclooxygenase-2 [155]. In diabetic mouse models, treatment with EGCG significantly decreased the expressions of various inflammatory markers (such as vascular cell adhesion molecule-1 (VCAM-1) and ICAM-1) and improved renal pathological alterations (such as glomerular enlargement and mesangial matrix expansion) [156]. Nonetheless, these protective properties of EGCG were not observed in the *Nrf2*-knockout diabetic mouse models, which is indicating that the anti-inflammatory properties of EGCG were facilitated via the Nrf2 signaling pathway [156].

### 4.3. Resveratrol

Resveratrol is a polyphenolic compound naturally present in numerous fruits and vegetables including peanuts, grapes, and peanut sprouts [146]. Resveratrol exerts anti-carcinogenic, anti-inflammatory, and antioxidant effects [171]. Resveratrol shows neuroprotective effects through its role in OS pathways [172,173]. It has been confirmed that resveratrol decreases various inflammatory mediators in the case of DN. Resveratrol also reduced neuropathy-related neuroinflammation via suppressing NF-κB to down-regulate the expressions of several proinflammatory factors, including iNOS, IL-6, and TNF-α [174]. Resveratrol ameliorated DN-linked sensorimotor disruptions by suppressing the activities of nicotinamide adenine dinucleotide phosphate oxidase TNF-α [158,175,176,177,178]. Nonetheless, clinical studies are required to demonstrate the aforesaid effects of resveratrol.

Resveratrol also has the potential to delay or even prevent the onset and development of DR. It decreases the advancement of DR via downregulating neuronal apoptosis, suppressing inflammation, and inhibiting angiogenesis [179,180,181,182]. Resveratrol also has the ability to counter OS [183]. Resveratrol blocked endothelial hyperpermeability and vascular lesions, which can further result in the loss of pericytes and capillary leakage [184,185]. Resveratrol-mediated VEGF suppression also caused the reduction of major mediators and the initiators of DR [186]. In a study, Xian et al. [157] observed that treatment with resveratrol markedly decreased blood glucose levels within 28 days of the study; however, the hypoglycemic activity did not last long. Moreover, resveratrol decreased the expressions of multiple inflammatory factors (such as NADPH Oxidase 4 (NOX4), NF-_K_B (P65), and RAGE), 24 h urinary microalbumin quantitative, serum creatinine (SCr), and blood urea nitrogen level, and ameliorated renal pathological structure. Therefore, resveratrol ameliorates renal functions by ameliorating the metabolic memory of hyperglycemia and anti-inflammatory property [157].

Resveratrol also decreased OS and ameliorated renal function in T1DM rat models [173]. Treatment with resveratrol markedly reduced protein carbonyl OS markers and superoxide anions [158]. In diabetic mouse models, resveratrol decreased mesangial cell glucotoxicity and renal lipotoxicity via activating PPARγ co-activator 1α [187]. Moreover, resveratrol decreased the IL-1β levels in STZ-induced diabetic rat kidneys, along with a marked rise in the levels of IL-6 and TNF-α (independent of activation of NF-κB), which further indicates that resveratrol exerts both suppressing and inducing actions on cytokines concurrently, and therefore attaining the optimal dose might be important to achieve efficacy [158,188].

### 4.4. Genistein

Genistein is an isoflavone found in various commonly consumed vegetables, legumes, and soy products [147]. Genistein exerts potent anti-inflammatory functions via suppressing multiple signaling pathways [189]. In a study, Li et al. [159] observed that genistein improved kidney functions as well as decreased levels of blood glucose and SCr. In addition, these researchers observed the downregulated expressions of p53, p65, MAPK, and NOX4. Genistein also elevated mitochondrial membrane potential, protected podocyte integrity, and decreased OS and expansion of the mesangial matrix. Therefore, this isoflavone may alleviate DNP by exerting an anti-inflammatory effect, improving mitochondrial function, and suppressing the MAPK/NF-κB signaling pathway [159]. In a different study, Ibrahim et al. [160] reported that genistein has the ability to inhibit TNF-α secretion and the phosphorylation of P38 and ERK in activated microglial cells, via suppressing tyrosine kinase. Thus, genistein might have the capacity in reducing diabetes-mediated inflammation in the retina by interfering with various inflammatory signaling pathways (including P38 MAPKs and ERK) that occurs in activated microglia. Collectively, these findings suggest that genistein might have the potential to control early pathological mechanisms prior to the incidence of vision loss among diabetic patients [160].

In a study, Ibrahim et al. [160] observed that the intravitreal injection of genistein markedly suppressed TNF-α mRNA and protein concentrations in the diabetic retina, which further suggests the therapeutic effect of genistein on inflammation in the STZ-diabetic model. In addition, genistein averted retinal microglia from upregulating IBA1 (*ionized calcium-binding adapter molecule 1)* mRNA, which indicates that genistein decreases the retinal inflammatory responses and the expressions of inflammatory cytokines via weakening activation of microglia [160]. On the other hand, the ability of genistein to suppress the activities of tyrosine kinases may be due to the antioxidant and anti-inflammatory effects of genistein [190]. ROS generation dependent on glycated albumin-induced NADPH oxidase may change the function of tyrosine phosphatase, which usually antagonizes the functions of tyrosine kinases [191]. In addition to this, expressions of tyrosine kinase functions are unimpeded, which can lead to the buildup of phosphorylated tyrosine residues. Thus, the capacity of genistein to decrease the phosphorylation of tyrosine might be due to the blockade of the inactivation of tyrosine phosphatase through its antioxidant function. Moreover, changes in the functions of transcription factors associated with inflammation including NF-κB was found to be a crucial constituent of NADPH oxidase-dependent redox signaling [192]. NF-κB is a pleiotropic mediator of numerous proinflammatory cytokines that are activated via many stimuli including diabetic stress [193]. Collectively, these findings suggest that genistein suppresses the activation of NF-κB in various cell types under stress conditions, which is further indicating that the capacity of genistein to decrease the albumin-induced TNF-α release might be owing to the activation of NF-κB [160,194,195].

### 4.5. Berberine

Berberine, a naturally occurring compound isolated from *Hydrastis canadensis* and *Coptis chinensis,* has anti-inflammatory and antioxidant properties [196]. The apoptosis of Müller cells can affect retinal neurons and vasculature, which can result in retinal complications of DM including DR [197]. In a study, Zhai et al. [161] observed that berberine deactivated the NF-κB signaling, decreased diabetes-mediated OS, and reduced the apoptosis of the retinal ganglion cell layer in a rat model. Berberine also reversed HG-mediated effects on OS and apoptosis in Müller cells in vitro via deactivating the NF-κB signaling. These findings indicate that berberine has the capacity to provide protection against DR via suppressing cell apoptosis and decreasing OS via the deactivation of the NF-κB signaling [161].

In a different study, Zan et al. [162] observed that treatment with berberine (20, 60 mg/kg; 30, 90 mg/kg) inhibited the expressions of TRPV1 and PKCε might be activated via hyperglycemia-induced inflammation. These researchers also demonstrated its ability to decrease TNF-α overexpression in diabetic mouse and rat models. TNF-α is directly linked with diabetic peripheral neuropathy. In addition, they summarized that berberine exerted its therapeutic actions by blocking the PKC pathway and inhibiting the inflammatory pathway to suppress the activation of TRPV1, which can result in neuronal injury and diabetic pain [162].

### 4.6. Quercetin

Quercetin is a potent natural flavonoid found widely in citrus fruits, broccoli, cherries, berries, grapes, and onions [149]. It has been observed that treatment with quercetin might provide immunoprotective, antiviral, anti-inflammatory, and antioxidant effects [198,199,200]. In a study, Zhao et al. [163] reported that quercetin ameliorated pathological alterations, nerve conduction velocity, and the mechanical withdrawal threshold in the sciatic nerves of DPN rat models. Furthermore, quercetin markedly improved the rise in the expressions of IL-1β and TNF-α. The administration of quercetin at a high dose markedly decreased the expressions of NF-κB, myeloid differentiation factor 88, and TLR4 in the sciatic nerves of DPN rat models. It has also been observed that quercetin did not reverse deceased body weight or reduce the blood glucose level, however it exerted neuroprotective and anti-inflammatory effects, which can be beneficial in DPN treatment [163].

In a different study, Hu et al. [201] observed that quercetin markedly elevated the functions of catalase (CAT) and superoxide dismutase. Quercetin also considerably reduced the levels of IL-1β, TNF-α, malondialdehyde, urine albumin, urine protein, blood urea nitrogen, SCr, and renal index. Moreover, quercetin decreased renal OS and inflammation, as well as ameliorated the renal function in DNP animal models.

### 4.7. Naringin

Naringin is a flavonoid glycoside abundantly found in the skin of oranges and grapefruit [150]. It has been reported that naringin has the ability to suppress OS and inflammation in diabetes [164]. In a study, Liu et al. [164] observed that naringin showed protective effects on retinal tissues of HG-induced rMC-1 cell line in vitro and STZ-induced rat *model* in vivo. This flavonoid glycoside also decreased DR in vitro and in vivo by suppressing OS and inflammation. Naringin also decreased DR by suppressing NF-κB signaling. However, more studies are required to demonstrate the capacity of naringin as an effective and novel therapy for DR [164]. In another study, Tsai et al. [202] observed that administration of naringenin at 2% reduced expressions of TGF-β1, fibronectin, and type IV collagen. In addition, naringenin at 1% and 2% decreased the functions of PKC and inhibited protein generation, mRNA expression, and NF-κB p65 function in the kidney. Nonetheless, only at 2%, naringenin decreased protein generation, mRNA expression, and NF-κB p50 function. These findings suggest that naringenin may decrease DR through its anti-fibrotic and anti-inflammatory effects [202]. In STZ-induced DNP rat models, Zhang et al. [165] showed that naringin decreased OS and kidney damage as well as inhibited ROS and apoptosis in HG-induced podocytes. Naringin also suppressed the expressions of NOX4 in HG-induced podocytes and STZ-induced DNP rat models. NOX4 downregulation reduced ROS levels and apoptosis in HG-induced podocytes. Collectively, these results indicate that naringin shows its protective properties in DNP via suppressing NOX4 expression [165].

### 4.8. Apigenin

Apigenin is a naturally occurring flavonoid with anti-fibrotic and anti-inflammatory effects. It has been reported that this flavonoid exerts a protective effect in the case of DNP [166]. Apigenin is widely found in plant-derived beverages, common fruits, and various vegetables including wheat sprouts, chamomile, tea, oranges, onions, and parsley [151]. In a study, Hou et al. [166] reported that the protective action of apigenin was linked with the expression regulations of miR-423-5p-USF2 axis, as confirmed by the observation that apigenin may ameliorate aberrantly up-regulated USF2 and down-regulated miR-423-5p in an in vitro DNP model. Furthermore, miR-423-5p was markedly decreased in the kidney tissues of DNP individuals. It was also observed that miR-423-5p up-regulation may ameliorate HG-induced cell damage by suppressing inflammation and apoptosis, as well as ameliorating podocyte vitality, and this protective action was linked with its specific inhibition of ROS and Nox4 functions [203].

### 4.9. Kaempferol, Baicalein, and Eriodictyol

Kaempferol is a polyphenol present in vegetables and fruits [204]. Kaempferol was found to exert anti-inflammatory effects in both in vitro and in vivo settings. It also ameliorated various inflammation markers [205]. Sharma et al. [206] showed that this polyphenol has the capacity to suppress fibrosis (expressions of extracellular matrix protein and TGF-β1), several proinflammatory CKs (IL-1β and TNF-α), OS, and hyperglycemia-induced RhoA activation in both RPTEC and NRK-52E RPTEC cells. Collectively, these findings suggest that kaempferol might be useful in DNP treatment [206].

Baicalein is a flavonoid present in *Scutellaria baicalensis (a Chinese herb)* [207]. *It has both* anti-apoptotic and anti-inflammatory properties. In addition, baicalin has the ability to protect the kidney and decrease insulin resistance [208]. Persistent hyperglycemia can result in inflammation and OS in DM, which can further result in the development and advancement of DNP. Baicalein decreased inflammation and OS in DNP via modulating MAPK and Nrf2 signaling. Therefore, the administration of baicalein might be effective in DNP treatment [167].

Eriodictyol is a flavonoid found in numerous vegetables and fruits, which has both antioxidant and anti-inflammatory properties [209,210]. Eriodictyol was found to decrease early retinal and plasma irregularities in STZ-induced diabetic rat models [168]. Eriodictyol also exerted antioxidant and anti-inflammatory effects via controlling Nrf2 signaling [209,211]. In a study, Lv et al. [212] observed that eriodictyol inhibited HG-induced apoptosis, inflammation, and OS in RGC-5 cells. It also enhanced Nrf2/HO-1 pathway activation in the HG-induced RGC-5 cell line. Collectively, these results indicate that the capacity of eriodictyol in protecting the RGC-5 cells against HG-induced damage via controlling the Nrf2/HO-1 signaling pathway could be effective in DR management [212].

## 5. Future Perspective

Currently, there is no available safe and effective therapeutic agent to treat DN [213]. Since bioactive compounds show effective anti-inflammatory and neuroprotective effects, therefore they can be considered in the management of various diabetes-linked neurological problems, including DN. On the other hand, bioactive compounds can regulate DR via multiple signaling pathways. They can also be used as an effective and safe therapy to reverse diabetes-associated impairment of vision. Therefore, more preclinical, and clinical studies are required to evaluate the effectiveness of these bioactive compounds in regulating various signaling mechanisms. Furthermore, quantitative structure-activity relationship (QSAR) modeling can also be used to synthesize potent derivatives of natural drugs [214]. There are no available therapeutic agents to improve DR-related neurodegenerative alterations in the retina.

Thus, the development and usage of safe and effective bioactive compounds from plants can be economically cheaper alternatives to treat DR. Moreover, many bioactive compounds have antidiabetic effects as well as the potential to increase neurotrophic effects, reduce OS, and decrease apoptosis in the retina of diabetic rat models. More studies are also required to obtain better knowledge regarding the phytochemicals-mediated reduction of microvascular complications of diabetes, including DR. Clinical trials are also needed to demonstrate the neuroprotective, antioxidant, and antidiabetic properties of bioactive compounds in patients with DR [215]. Bioactive compounds that alleviate DNP via controlling various signaling mechanisms can be further developed as a safe and effective therapy in reversing the loss of renal functions. Bioactive compounds derived from plants that have the potential to regulate AKT, mTOR, adenosine monophosphate-activated protein kinase (AMPK), AGEs, growth factors, proinflammatory CKs, OS, and various other signaling mechanisms associated with DNP, might prove to be an effective therapy to treat DNP. However, more studies are needed to demonstrate the potential of bioactive compounds in preclinical animal models of DNP. Moreover, novel, potent, and anti-inflammatory bioactive compounds can also be synthesized by QSAR modeling, based on their chemical structures [216].

## 6. Conclusions

The aforesaid findings summarized in this article clearly demonstrate that bioactive compounds derived from plants are effective in improving vision and inhibiting inflammation-mediated cellular injury in the retina. In daily diets, the addition of medicinal plants can improve human health and decrease the risk of diabetes and its associated microvascular complications including DR and DNP. Moreover, bioactive compounds can act as effective natural reservoirs of glycemic control and exert potent neuroprotective properties, which might be effective in managing DN.

## Figures and Tables

**Figure 1 molecules-27-07352-f001:**
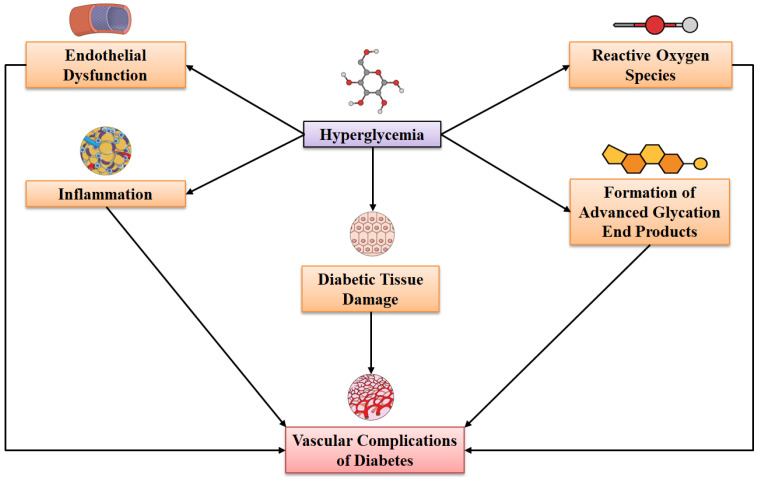
Pathogenic mechanisms that are linked with the development of vascular complications of diabetes mellitus.

**Figure 2 molecules-27-07352-f002:**
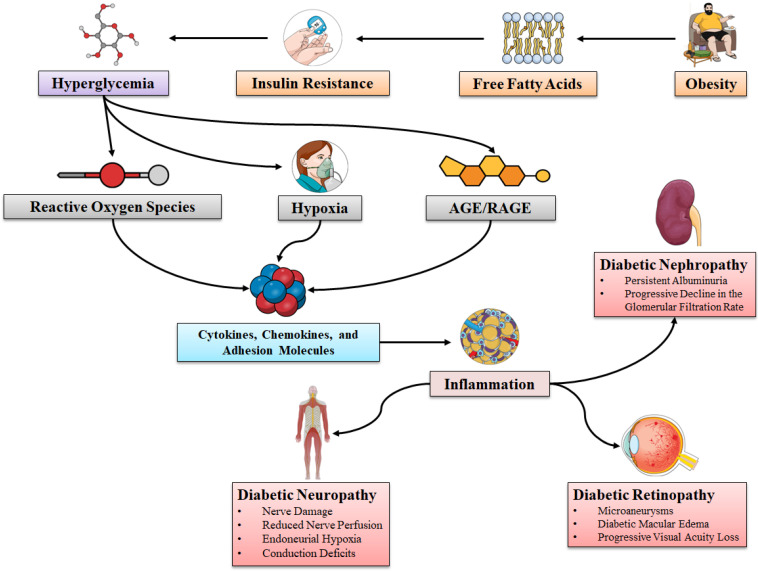
The role of inflammation in diabetic neuropathy, retinopathy, and nephropathy. (Abbreviations: AGEs, advanced glycation end products; RAGE, receptor for advanced glycation endproducts).

**Figure 3 molecules-27-07352-f003:**
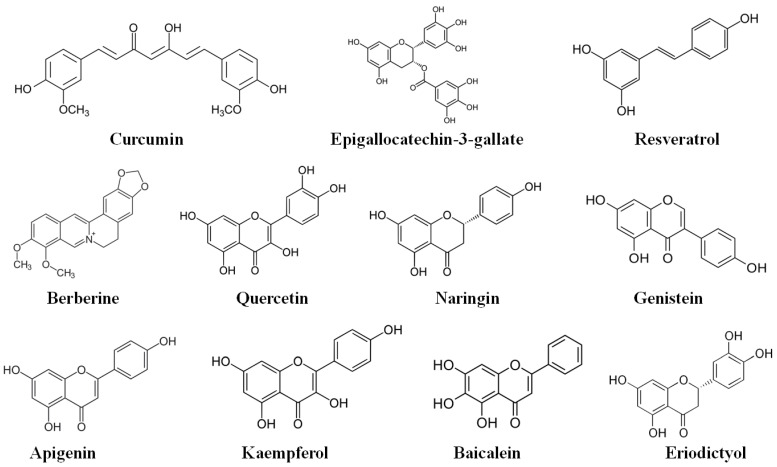
Chemical structures of bioactive compounds that have the potential to treat inflammation in microvascular complications of diabetes.

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
