# Peer review of "Potential Roles of Anti-Inflammatory Plant-Derived Bioactive Compounds Targeting Inflammation in Microvascular Complications of Diabetes"

_molecules, 2022, doi:10.3390/molecules27217352_

Round 1

Reviewer 1 Report

   1.This article provides a comprehensive overview of diabetes inflammatory molecules and potential bioactive molecules.

   2.In this paper, the diabetes related molecules only made a summary, the depth of thinking

Author Response

Thank you for the favourable comments. I am happy to see your observation about the strength of my manuscript. 

Reviewer 2 Report

In MS entitled “Role of Inflammation in Microvascular Complications of Diabetes: From Molecular Pathogenesis to Potential Bioactive Molecular Pathogenesis to Potential Bioactive Molecules” Kaabi reviewed role of inflammation in microvascular complications of diabetes. In first part of this review author provided an overview of role of inflammation in diabetes and its complications. The weakness of this part of the paper is the originality of the topic, since there are few reviews with the same or very similar topic:

1.       Nguyen, D. V., Shaw, L. C., & Grant, M. B. (2012). Inflammation in the pathogenesis of microvascular complications in diabetes. Frontiers in endocrinology3, 170.

2.       Faselis, C., Katsimardou, A., Imprialos, K., Deligkaris, P., Kallistratos, M., & Dimitriadis, K. (2020). Microvascular complications of type 2 diabetes mellitus. Current vascular pharmacology18(2), 117-124.

3.       Tsalamandris, S., Antonopoulos, A. S., Oikonomou, E., Papamikroulis, G. A., Vogiatzi, G., Papaioannou, S., ... & Tousoulis, D. (2019). The role of inflammation in diabetes: current concepts and future perspectives. European cardiology review14(1), 50.

In the second part of the review author analyzed therapeutic effects of bioactive agents in treatment of microvascular complications of diabetes. Considering the lack of originality and novelty of first part of this review, I suggest that the author prepare a mini-review in which effects of bioactive agents in treatment of microvascular complications of diabetes will be analyzed, and resubmit it to Molecules.

Other comments:

In title it should be emphasized that bioactive molecules have the role in the treatment of microvascular complications.

In Introduction “we” should be changed / deleted since there is only one author of this MS.

In Introduction percentage should be separated from numerical value.

English, both style and grammar, should be corrected.

Author Response

Dear Reviewer

Please see the attached reply to the comments

Reviewer 1

In MS entitled “Role of Inflammation in Microvascular Complications of Diabetes: From Molecular Pathogenesis to Potential Bioactive Molecular Pathogenesis to Potential Bioactive Molecules” Kaabi reviewed role of inflammation in microvascular complications of diabetes. In first part of this review author provided an overview of role of inflammation in diabetes and its complications. The weakness of this part of the paper is the originality of the topic, since there are few reviews with the same or very similar topic:

  1. Nguyen, D. V., Shaw, L. C., & Grant, M. B. (2012). Inflammation in the pathogenesis of microvascular complications in diabetes. Frontiers in endocrinology3, 170.

Reply: My manuscript is significantly different from this review. In addition, the above review covers only the role of inflammation in the pathogenesis of microvascular complications in diabetes, whereas my manuscript extensively covers the general role of inflammation in diabetes mellitus and therapeutic potential of anti-inflammatory bioactive agents in the treatment of microvascular complications of diabetes.           

  1. Faselis, C., Katsimardou, A., Imprialos, K., Deligkaris, P., Kallistratos, M., & Dimitriadis, K. (2020). Microvascular complications of type 2 diabetes mellitus. Current vascular pharmacology18(2), 117-124.

Reply: Above mentioned review covers only the general overview of microvascular complications of type 2 diabetes mellitus. My manuscript is considerably different from this manuscript, since my manuscript is particularly focusing on the role of inflammation in the pathogenesis of microvascular complications of diabetes and the treatment options of these complications with potential bioactive agents.

  1. Tsalamandris, S., Antonopoulos, A. S., Oikonomou, E., Papamikroulis, G. A., Vogiatzi, G., Papaioannou, S., ... & Tousoulis, D. (2019). The role of inflammation in diabetes: current concepts and future perspectives. European cardiology review14(1), 50.

Reply: Aforementioned review covers solely the general overview of role of inflammation in diabetes mellitus, while my manuscript has focused mainly on the role of inflammation in the pathogenesis of microvascular complications of diabetes. Moreover, the above mentioned review briefly discussed the potential of synthetic drugs to treat inflammation in diabetes mellitus, while I have extensively covered the therapeutic potential of anti-inflammatory bioactive agents in the treatment of microvascular complications of diabetes. Therefore, my manuscript is markedly different from this review.             

In the second part of the review author analyzed therapeutic effects of bioactive agents in treatment of microvascular complications of diabetes. Considering the lack of originality and novelty of first part of this review, I suggest that the author prepare a mini-review in which effects of bioactive agents in treatment of microvascular complications of diabetes will be analyzed, and resubmit it to Molecules.

Reply: My above justifications clearly denote that my manuscript has originality and novelty as well as its usefulness. However, I have changed the title in the revised version, so that it more prominently reflects the potential roles of anti-Inflammatory bioactive compounds in alleviating inflammation in microvascular complica-tions of diabetes.

Other comments:

In title it should be emphasized that bioactive molecules have the role in the treatment of microvascular complications.

Reply: I have changed the title in the revised version, so that it more prominently reflects the potential roles of anti-Inflammatory bioactive compounds in alleviating inflammation in microvascular complications of diabetes.

In Introduction “we” should be changed / deleted since there is only one author of this MS.

Reply: I have changed “we” to “I” in the introduction and abstract sections in the revised version.

In Introduction percentage should be separated from numerical value.

Reply: All the percentage symbols have been replaced with the word “percent” in the revised version.

English, both style and grammar, should be corrected.

Reply: I have thoroughly checked the manuscript again for such errors, where found I have corrected them. 

Reviewer 3 Report

In this review, the author presented the inflammatory effects due to the diabetes causing various physiological complications like diabetic nephropathy (DNP), diabetic neuropathy (DN), and diabetic retinopathy (DR) and summarizes the potential uses of various anti-inflatory agents (plant based) for their remedies.

My comments are as follows:

1. The title of the manuscript is correctly presented

2. The abstract presented flawlessly which represents the manuscript rationally. No typos were observed.

3. In the introduction section, the authors gave us a brief idea about Diabetes mellitus (DM) and its inflammatory effects which leads us to diabetic nephropathy (DNP), diabetic neuropathy (DN), and diabetic retinopathy (DR) in a proper schematic representation.

     a) In Figure 1, a reference may be added which looks more rational.

     b) In Section 2 may be merged with introduction section.

4. In section 3, the authors described the role of Inflammation in Microvascular Complications of Diabetes in separate sections thoroughly with authentic references. For all three cases like diabetic nephropathy (DNP), diabetic neuropathy (DN), and diabetic retinopathy (DR); it is advisable to incorporate schematic diagram with proper mechanistic pathways for better understanding.

5. In section 4, the author presented the Therapeutic Potential 11 different of Anti-Inflammatory Bioactive Agents (plant based small molecules) to reduce DR, DN, DNP etc.  thoroughly.

a) All molecules are found to be naturally abundant and reports showed that these could be promising therapeutic alternatives to treat T2DM.

b) Any side-effects of these molecules are reported so far?

b) Applications are limited only for animal models or is there any clinical developments happening for these molecules?

c) The author should show a comparison between these of Anti-Inflammatory Bioactive Agents and currently used antidiabetic drugs like metformin, GLP1 agonists, insulin derivatives in terms of anti-inflammatory properties.

6. It is recommended to emphasis the importance of iminosugars and sugar derivatives as an anti-diabetic agents, and it is suggested to cite following relevant articles related to iminosugars in introduction section.

  1. Nash, R. J.; Kato, A.; Yu, C-. Y.; Fleet, G. W. J. Iminosugars as therapeutic agents: recent advances and promising trends. Future Med. Chem. 2011, 3, 1513−1521.
  2. Yang, L.-F.; Shimadate, Y.; Kato, A.; Li, Y.-X.; Jia, Y.-M.; Fleet, G.W.J.; Yu, C.-Y. Synthesis and glycosidase inhibition of N-substituted derivatives of DIM. Org. Biomol. Chem. 2020, 18, 999–1011.
  3. Chennaiah, A.; Dahiya, A.; Dubbu, S.; Vankar, Y. D. A Stereoselective Synthesis of an Imino Glycal: Application in the Synthesis of (−)-1-Epi -Adenophorine and a Homoiminosugar. Eur. J. Org. Chem. 2018, 6574−6581.
  4. Chennaiah, A.; Bhowmick, S.; Vankar, Y. D. Conversion of glycals into vicinal-1,2-diazides and 1,2-(or 2,1)-azidoacetates using hypervalent iodine reagents and Me3SiN3. Application in the synthesis of N-glycopeptides, pseudo-trisaccharides and an iminosugar. RSC Adv. 2017, 7, 41755−41762.
  5. Rajasekaran, P.; Ande, C.; Vankar, Y. D. Synthesis of (5,6 & 6,6)-oxa-oxa annulated sugars as glycosidase inhibitors from 2-formyl galactal using iodocyclization as a key step. ARKIVOC 2022, vi, 5−23.

Overall, after addressing the points mentioned above, I recommend this review to publish in molecules.

Author Response

Dear Reviewer

Please see the attached reply to the comments

Reviewer 2

In this review, the author presented the inflammatory effects due to the diabetes causing various physiological complications like diabetic nephropathy (DNP), diabetic neuropathy (DN), and diabetic retinopathy (DR) and summarizes the potential uses of various anti-inflatory agents (plant based) for their remedies.

My comments are as follows:

  1. The title of the manuscript is correctly presented
  2. The abstract presented flawlessly which represents the manuscript rationally. No typos were observed.
  3. In the introduction section, the authors gave us a brief idea about Diabetes mellitus (DM) and its inflammatory effects which leads us to diabetic nephropathy (DNP), diabetic neuropathy (DN), and diabetic retinopathy (DR) in a proper schematic representation.
  4. a) In Figure 1, a reference may be added which looks more rational.

Reply: Figure 1 is original and made by me, therefore no reference was added.

  1. b) In Section 2 may be merged with introduction section.

Reply: Merging section 2 with the introduction will make the introduction section pretty big and difficult to follow by the readers. In addition, a separate section on “The Role of Inflammation in Diabetes Mellitus” will help the readers to understand and appreciate the role of inflammation.

  1. In section 3, the authors described the role of Inflammation in Microvascular Complications of Diabetes in separate sections thoroughly with authentic references. For all three cases like diabetic nephropathy (DNP), diabetic neuropathy (DN), and diabetic retinopathy (DR); it is advisable to incorporate schematic diagram with proper mechanistic pathways for better understanding.

Reply: Figure 2 already clearly summarizes the role of inflammation in diabetic neuropathy, retinopathy, and nephropathy. Moreover, the figure for each complication will make section 3 unnecessarily long.

  1. In section 4, the author presented the Therapeutic Potential 11 different of Anti-Inflammatory Bioactive Agents (plant based small molecules) to reduce DR, DN, DNP etc. thoroughly.
  2. a) All molecules are found to be naturally abundant and reports showed that these could be promising therapeutic alternatives to treat T2DM.
  3. b) Any side-effects of these molecules are reported so far?

Reply: No side effects related to the molecules were reported in the mentioned studies.

  1. b) Applications are limited only for animal models or is there any clinical developments happening for these molecules?

Reply: Clinical trials are yet to be carried out with the bioactive agents targeting inflammation in microvascular complications of diabetes. I have also mentioned this suggestion in the future perspective section. 

  1. c) The author should show a comparison between these of Anti-Inflammatory Bioactive Agents and currently used antidiabetic drugs like metformin, GLP1 agonists, insulin derivatives in terms of anti-inflammatory properties.

Reply: Metformin, GLP1 agonists, and insulin derivatives are synthetic in nature, whereas the focus of my manuscript is plant-derived bioactive compounds. Nonetheless, a general comparison between these agents and plant-derived bioactive compounds has been provided in the introduction section.

  1. It is recommended to emphasis the importance of iminosugars and sugar derivatives as an anti-diabetic agents, and it is suggested to cite following relevant articles related to iminosugars in introduction section.
  1. Nash, R. J.; Kato, A.; Yu, C-. Y.; Fleet, G. W. J. Iminosugars as therapeutic agents: recent advances and promising trends. Future Med. Chem. 2011, 3, 1513−1521.
  2. Yang, L.-F.; Shimadate, Y.; Kato, A.; Li, Y.-X.; Jia, Y.-M.; Fleet, G.W.J.; Yu, C.-Y. Synthesis and glycosidase inhibition of N-substituted derivatives of DIM. Org. Biomol. Chem. 2020, 18, 999–1011.
  3. Chennaiah, A.; Dahiya, A.; Dubbu, S.; Vankar, Y. D. A Stereoselective Synthesis of an Imino Glycal: Application in the Synthesis of (−)-1-Epi -Adenophorine and a Homoiminosugar. Eur. J. Org. Chem. 2018, 6574−6581.
  4. Chennaiah, A.; Bhowmick, S.; Vankar, Y. D. Conversion of glycals into vicinal-1,2-diazides and 1,2-(or 2,1)-azidoacetates using hypervalent iodine reagents and Me3SiN3. Application in the synthesis of N-glycopeptides, pseudo-trisaccharides and an iminosugar. RSC Adv. 2017, 7, 41755−41762.
  5. Rajasekaran, P.; Ande, C.; Vankar, Y. D. Synthesis of (5,6 & 6,6)-oxa-oxa annulated sugars as glycosidase inhibitors from 2-formyl galactal using iodocyclization as a key step. ARKIVOC 2022, vi, 5−23.

Reply: The suggested articles- 1 and 3 are relevant and go with the scope of my article. These articles have been cited in the introduction section. However, other suggested articles are not relevant to my manuscript. 

Round 2

Reviewer 2 Report

Author replied to some of the objections. However, on my comment "In Introduction percentage should be separated from numerical value." The author replied 

" All the percentage symbols have been replaced with the word “percent” in the revised version." However, percentage symbols should not be replaced with the word “percent”, but percentage symbol should be separated from numerical value. Instead of using "I" in the MS, it is better to apply passive tense. English language and style of the MS should be corrected. 

Author Response

Author replied to some of the objections. However, on my comment "In Introduction percentage should be separated from numerical value." The author replied 

" All the percentage symbols have been replaced with the word “percent” in the revised version." However, percentage symbols should not be replaced with the word “percent”, but percentage symbol should be separated from numerical value.

Reply: All the percentages have been separated from numerical values.

Instead of using "I" in the MS, it is better to apply passive tense.

Reply: All the sentences containing "I" in the manuscript have been rewritten in passive forms.

English language and style of the MS should be corrected. 

Reply: Entire manuscript has been revised again to find any error in English language/sentence structure. Where detected, I have corrected those errors. I have prepared the manuscript by following the template/guidelines of MDPI and did not find any deviations. However, if any specific errors are pointed, I would be more than happy to correct them. 
